# Influence of Sandblasting Particle Size and Pressure on Resin Bonding Durability to Zirconia: A Residual Stress Study

**DOI:** 10.3390/ma13245629

**Published:** 2020-12-10

**Authors:** Sung-Min Kwon, Bong Ki Min, Young Kyung Kim, Tae-Yub Kwon

**Affiliations:** 1Department of Medical & Biological Engineering, Graduate School, Kyungpook National University, 2177 Dalgubeol-daero, Jung-gu, Daegu 41940, Korea; sungmin@knu.ac.kr; 2Center for Research Facilities, Yeungnam University, 280 Daehak-ro, Gyeongsan 38541, Korea; 3Department of Conservative Dentistry, School of Dentistry, Kyungpook National University, 2177 Dalgubeol-daero, Jung-gu, Daegu 41940, Korea; wisekim@knu.ac.kr; 4Department of Dental Biomaterials, School of Dentistry and Institute for Biomaterials Research & Development, Kyungpook National University, 2177 Dalgubeol-daero, Jung-gu, Daegu 41940, Korea

**Keywords:** zirconia, sandblasting, resin bonding, residual stress

## Abstract

The influence of residual stress induced by sandblasting the zirconia ceramic surface on the resin bonding to the ceramic is still unclear. The effect of four different sandblasting conditions (with 50 and 110 μm alumina at pressures of 0.2 and 0.4 MPa) on the bonding of adhesive resin cement (Panavia F 2.0) to zirconia (Cercon^®^ ht) was investigated in terms of residual stress. The surface roughness and water contact angle of the zirconia surfaces were measured. The tetragonal-to-monoclinic (t–m) phase transformation and residual stresses (sin^2^*ψ* method) were studied by X-ray diffraction. The resin-bonded zirconia specimens were subjected to shear bond strength (SBS) tests before and after thermocycling (10,000 and 30,000 cycles) (*n* = 10). As the particle size and pressure increased, the roughness gradually and significantly increased (*p* = 0.023). However, there were no significant differences in roughness-corrected contact angle among all the sandblasted groups (*p* > 0.05). As the particle size and pressure increased, the m-phase/(t-phase + m-phase) ratios and compressive residual stresses gradually increased. After thermocycling, there were no significant differences in SBS among the sandblasted zirconia groups (*p* > 0.05). In conclusion, increased surface roughness and residual stress do not directly affect the resin bonding durability.

## 1. Introduction

In dentistry, the introduction of computer-aided design/computer-aided manufacture (CAD/CAM) systems significantly increased the use of all-ceramic restorations [1]. Zirconia is one of the primary reinforced ceramic substrates used in the CAD/CAM process [1]. This ceramic is in popular use mainly due to its exceptional mechanical properties, as well as its biocompatibility and resistance to corrosion [1,2,3], although its limited esthetics make it difficult to match the restorations with existing dentition [3,4].

Unlike silica-based ceramics, zirconia is a silica-free polycrystalline ceramic, which makes it resistant to traditional hydrofluoric acid etching and silane coupling agent treatments [5]. The bond strength and durability of various bonding methods to zirconia, including several innovative ceramic surface treatments, have been investigated. However, a standardized bonding protocol for zirconia-based restorations has not yet been established [5]. It has been shown that sandblasting pretreatment results in relatively high bond strengths to zirconia when combined with resin materials containing phosphate monomers, such as 10-methacryloyloxydecyl dihydrogen phosphate (MDP) [6,7]. This mechanical surface conditioning can enhance resin–zirconia adhesion by cleaning the ceramic surface, increasing the roughness and bonding area, and enhancing the wetting of resin materials [7]. Sandblasting with silica-coated alumina particles (tribochemical silica-coating or silicatization) to improve the resin bonding with zirconia is another method used in dentistry [8,9]. The presence of silica on the surface is a prerequisite for the formation of durable siloxane bonding [8]. A high blasting pressure can result in the superior attachment of silica-coated alumina particles on the zirconia surface [9]. However, the silica particles do not deeply penetrate into the sandblasted zirconia and only loosely cover the surface [8,9].

Sandblasting involves impacting the substrate surface with hard particles at high velocities, thereby eroding the material and leaving a roughened surface with high wettability [10]. In the case of zirconia ceramic, sandblasting induces undesirable tetragonal-to-monoclinic (t–m) phase transformation and microcrack formation at the near surface zone [10]. It also creates residual stress in the zirconia surface, mainly due to local irreversible deformation at the impact sites [10]. Chintapalli et al. [11] showed that the extent of subsurface t–m transformation and substrate damage caused by sandblasting were dependent on sandblasting conditions (abrasive particle size and pressure). They concluded that mild sandblasting was more beneficial than the harsh one since the former induced limited damage. Okada et al. [12] found that there was an optimal blasting pressure range to increase the zirconia flexural strength, although surface roughness increased along with blasting pressure.

Residual stress can affect the bond strength of the coating layer on a substrate [13]. Yang and Chang [13] reported that compressive residual stress weakened the adhesion at the interface of the hydroxyapatite (HA) coatings and the titanium. Similarly, the residual stress formed on the zirconia surface after sandblasting may affect the resin bond strength to zirconia. However, there are still relatively few studies available on the direct influence of residual stress induced by sandblasting the zirconia surface on the resin bonding to zirconia. The purpose of this study was to investigate the influence of different sandblasting conditions on the resin bonding to zirconia ceramic in terms of residual stress. The null hypotheses were that the different sandblastings would not result in (1) different residual stresses of the zirconia surfaces nor (2) different resin bond strengths.

## 2. Materials and Methods

### 2.1. Specimen Preparation

A total of 210 rectangle-shaped (10 × 10 × 2 mm) zirconia plates (Cercon^®^ ht, DeguDent, Hanau, Germany) were prepared by sintering following the manufacturer’s recommendations and polished up to a 1 μm diamond suspension. Excluding for the control (polished surface) group, sandblasting was carried out on the zirconia surfaces with two different sizes of alumina particles (Cobra, Renfert GmbH, Hilzingen, Germany) of 50 and 110 μm [9] and two different pressures of 0.2 and 0.4 MPa [10] (group codes: 50M-2B, 50M-4B, 110M-2B, and 110M-4B, respectively). The blasting nozzle tip was positioned perpendicular to the zirconia surface at a distance of 10 mm. The duration of the sandblasting was 10 s. The sandblasted surfaces were ultrasonically cleaned in isopropyl alcohol and then water for 10 min each.

### 2.2. SEM and EDS Analyses

The zirconia surfaces were examined after gold coating with scanning electron microscopy (SEM, JSM-6700F, Jeol, Tokyo, Japan) at a magnification of 1000× and an accelerating voltage of 10 kV (*n* = 1). In addition, the chemical composition of the surfaces was verified with energy dispersive X-ray spectroscopy (EDS, Oxford Instruments, High Wycombe, UK).

### 2.3. Surface Roughness and Contact Angle (CA) Measurements

The average roughness (*R*_a_) and average peak-to-valley height (*R*_z_) of the zirconia surfaces were measured with a profilometer (Surftest SV-402, Mitutoyo Corp., Tokyo, Japan). The stylus speed was 0.1 mm/s, the cutoff was 0.8 mm, and the range was 600 μm. The values for each zirconia specimen were recorded as the average of three readings (*n* = 5).

CAs of water droplet (5 μL volume) on the zirconia surfaces were measured with a CA goniometer (Attension Theta, Biolin Scientific Oy, Espoo, Finland) (*n* = 5). The CAs were subjected to roughness correction according to Wenzel’s equation, which states the relation between the measured CA (*θ*_m_) and Young’s CA (*θ*_Y_) [14]: cos*θ*_m_ = *r* cos*θ*_Y_, in which *r* denotes the ratio between the rough and the smooth surface areas of the specimen and can be calculated from the equation: *r* = 1 + *S*_dr_/100, where *S*_dr_ refers to the developed interfacial area ratio. The *S*_dr_ values of the zirconia surfaces were measured using a phase shift interferometer in the goniometer.

### 2.4. X-ray Diffraction (XRD) Analysis

The t–m phase transformation of the polished and sandblasted zirconia surfaces was investigated by XRD (D8 Discover, Bruker AXS GmbH, Karlsruhe, Germany) (*n* = 1). Spectra were obtained for 2*θ* between 25° and 100°. Cu *K*_α_ with wavelength *λ* = 1.5406 Å was employed as the radiation source.

XRD was also employed to measure the residual strains induced by polishing and sandblasting for known orientations relative to the zirconia surfaces. The sin^2^*ψ* method was used to calculate the residual stress from the strains in the zirconia specimens [13], in which *ψ* is the angle between the normal of the specimen and the normal of the diffracting plane. By altering the tilt of the zirconia specimen within the diffractometer, measurements of planes at an angle *ψ* were made. The diffractometer was set up to measure the strains perpendicular to the polishing/sandblasting direction—i.e., *φ* = 0°. Several XRD measurements were carried out at different *ψ* tilts of 0, 22.79, 33.21, 42.13, 50.77, 60.00, 71.67°, using the (002) reflection of the zirconia specimens. The diffraction angle 2*θ* = 62.5° was chosen and a linear elastic distortion of the crystal lattice plane was assumed. The stress producing the strain was calculated from the strain versus sin^2^*ψ* plots by considering the gradient of the line and the elastic properties of the zirconia, using the following equation [13]:(1)$$\sigma_{\varphi }\;=\;\left(\frac{E}{1+v}\right)\left(hkl\right)\frac{1}{d_{0}}{\left(\frac{\begin{array}{c} \partial d_{\varphi \psi } \end{array}}{\begin{array}{c} \partial \;\sin^{2}\psi \end{array}}\right)}_{\;}$$
where *σ_φ_* is the single stress acting in a chosen direction, *E* is the modulus of elasticity in MPa, *v* the Poisson’s ratio, (*hkl*) the Miller indices describing a family of crystalline planes, *d*_0_ the unstressed lattice spacing, and *φ* the angle between a fixed direction in the plane of the specimen and the projection in that plane of the normal of the diffracting plane. The lattice spacing *d_φψ_* is a linear function of sin^2^*ψ*.

### 2.5. Shear Bond Strength (SBS) Test

The polished (control) or sandblasted (50M-2B, 50M-4B, 110M-2B, and 110M-4B) zirconia specimens (*n* = 30 per sandblasting condition) were subdivided into three groups depending on the thermocycling conditions (*n* = 10). The adhesion procedure was carried out using the notched SBS test [15]. The zirconia ceramic surfaces for bonding were isolated using a bonding clamp containing a white plastic button mold with hole diameter of 2.38 mm (Ultradent Products Inc., South Jordan, UT, USA). An adhesive resin material (Panavia F 2.0, Kuraray Medical Inc., Okayama, Japan) was packed into the button mold insert and light-cured for 40 s using a dental light curing unit (Bluephase^®^ 20i, Ivoclar Vivadent, Schaan, Liechtenstein). The bonded specimen was then removed from the button mold. Only one resin cylinder was prepared on each zirconia specimen [15]. A third of all the bonded zirconia specimens were subjected to SBS test after storage in 37 °C water for 24 h. The remaining two-thirds were thermocycled either 10,000 or 30,000 times, prior to SBS test, between two water-baths of temperatures 5 °C and 55 °C, with a dwelling time of 30 s and an exchange time of 5 s between each bath, after water immersion at 37 °C for 7 days.

The resin-bonded zirconia specimens were engaged at their resin bases with a notched-edge shear blade in a universal testing machine (3343, Instron Inc., Canton, MA, USA) until failure occurred (crosshead speed = 1.0 mm/min) [15]. Bond strengths in MPa were obtained from the peak failure load divided by the adhesion area. After debonding, the zirconia surfaces were observed under an optical microscope (SZ61, Olympus, Tokyo, Japan) at a magnification of 30 × to determine the failure mode: A, interfacial adhesive failure; C, cohesive failure within the resin material; M, mixed failure.

### 2.6. Statistical Analysis

The surface roughness, CA, and SBS data did not meet the assumption of equal variances (Leven’s test). Thus, the Kruskal–Wallis test, followed by the Mann–Whitney test with significance level adjustment using the Benjamini and Hochberg method, was used for the statistical analysis of the data, using SPSS 17.0 for Windows (SPSS Inc., Chicago, IL, USA) at α = 0.05.

## 3. Results

Figure 1 shows the SEM images of the polished and sandblasted zirconia surfaces and corresponding EDS spectra. The SEM images show the smooth surface by polishing (a) and the roughened surfaces by sandblasting (b–e). The aluminum element was not detected by EDS on the polished surface. In contrast, the sandblasted zirconia surfaces showed a small amount (1.85–2.68 at%) of remaining aluminum even after ultrasonic cleaning.

The surface roughness values (*R*_a_ and *R*_z_) and water CAs (before and after roughness correction) for the polished and sandblasted zirconia specimens are summarized in Table 1. The polished surface exhibited very low *R*_a_ and *R*_z_ values. As the alumina particle size and pressure increased, both the *R*_a_ and *R*_z_ values gradually and significantly increased (*p* = 0.023). The measured CAs significantly decreased after sandblasting (*p* = 0.042) and gradually decreased by increasing the abrasive particle size and pressure. The 110M-4B group showed significantly lower CA than the 50M-2B and 50M-4B groups (*p* = 0.029). After roughness correction, however, there were no significant differences in CA among all the sandblasted groups (*p* > 0.05).

Figure 2 and Figure 3 show the XRD patterns of the polished and sandblasted zirconia surfaces. The spectrum of the polished specimen exhibited only the t-phase characteristic peaks (t (101), t (002), and t (110) peaks), without the appearance of the m-phase. After sandblasting, the m-phase peak (m (111)) newly appeared at an angle 2θ of 28.2°. The m-phase/(t-phase + m-phase) ratios gradually increased as the particle size and pressure increased.

The plots of strain versus sin^2^*ψ* for the polished and sandblasted zirconia surfaces and the residual stresses calculated from the linear dependence are shown in Figure 4 and Figure 5, respectively. The polished zirconia surface exhibited a slight residual stress (50 MPa). The larger alumina particle size and higher sandblasting pressure resulted in higher compressive residual stress (average values ranged from 451 to 905 MPa).

Table 2 summarizes the SBS values of the resin cement to the various zirconia ceramic surfaces before and after thermocycling. Regardless of the thermocycling condition, the polished zirconia group showed significantly lower SBS values than the sandblasted groups (*p* < 0.001). At 0 cycles, the 50M-4B and 110M-4B group values were significantly higher those of the 50M-2B and 110M-2B group (*p* < 0.05). At both 10,000 and 30,000 cycles, however, there were no significant differences in SBS among the sandblasted zirconia groups (*p* > 0.05). The polished zirconia surfaces showed only adhesive failure, irrespective of the thermocycling condition. Before thermocycling, the distribution of the failure modes for the sandblasted surfaces was 20–30% adhesive failures and 70–80% mixed failures. No cohesive failure within the resin cement was observed. After thermocycling, all sandblasted zirconia specimens exhibited exclusively adhesive failure after debonding.

## 4. Discussion

The purpose of the present study was to investigate the effect of four different sandblasting conditions (two blasting particle sizes and two pressures) on the resin bonding to zirconia ceramic in terms of residual stress. Based on the findings, the first null hypothesis, that the different sandblasting conditions would not result in different residual stresses of the zirconia surfaces, was rejected (Figure 5). The resin-bonded zirconia specimens were aged by thermocycling after a week of water saturation [15]. Thermocycling inducing the thermal stress and hydrolytic effect is widely used to evaluate the adhesion durability [16]. As an estimate of approximately 10,000 cycles per year is provisionally suggested [17], the 10,000 and 30,000 cycles used in this study thus represent approximately 1 and 3 years of clinical use, respectively. Before thermocycling, different SBS values were obtained by the different sandblastings (Table 2). However, no significant differences in SBS were detected among any of the sandblasting groups after thermocycling. Thus, the second null hypothesis was partially rejected.

It is known that the sandblasting of zirconia conditions the surface to increase the roughness, as well as to clean and activate the surface [7]. However, the alumina particles used for sandblasting may remain on the zirconia surface even after ultrasonic cleaning in isopropyl alcohol and water for 10 min each. As shown in Figure 1, only low levels of aluminum element were detected on all the ultrasonically cleaned zirconia surfaces after sandblasting although the detected amount gradually increased with greater abrasive particle size and pressure. Therefore, the presence of only a small amount of alumina particles on the zirconia surfaces does not seem to have impaired the resin bonding. In the case of the tribochemical silica coating, in contrast, the silica particles should remain even after ultrasonic cleaning to create durable siloxane bonding [8]. However, Lorente et al. [18] suggested that the silica particles uniformly cover the zirconia surface after the sandblasting, but the subsequent ultrasonic cleaning nearly completely removes the loosely bound particles. Therefore, the use of alumina particles at various blasting pressures might be preferred to that of silica-coated alumina particles [9].

The surface roughness values of the zirconia specimens were dependent on the sandblasting pressure as well as the alumina particle size (Table 1). Likewise, the measured CA values gradually decreased by increasing the particle size and pressure. The Young CA assumes that the surface is topographically smooth and chemically homogenous [14]. This is not true in the case of most real surfaces, especially roughened surfaces by sandblasting. In order to obtain the actual CAs, the CAs measured on the sandblasted zirconia surfaces were roughness-corrected using the *S*_dr_ values of the surfaces [14]. After roughness correction, the sandblasted surfaces did not exhibit significantly different CAs. This finding indicates that the more aggressive sandblasting treatments did not necessarily produce more chemically activated zirconia surfaces, the higher roughness values notwithstanding.

The XRD analysis showed that the m-phase newly formed on the sandblasted zirconia surfaces increased with an increase in particle size and blasting pressure (Figure 2 and Figure 3). The shape and diameter of abrasive particles are of primary importance in the sandblasting process [9]. The m-phase/(t-phase + m-phase) ratios increased by 83% and 44%, respectively, when the particle size was fixed and the blasting pressure was increased (50M-2B to 50M-4B and 110M-2B to 110M-4B). On the other hand, when the particle size was increased and the pressure was decreased (50M-4B to 110M-2B), the increase in the ratio was only 9%. These findings indicate that the formation of the m-phase induced by sandblasting depends more on the blasting pressure than the particle size. The polishing of the zirconia surface resulted in the creation of a slight compressive residual stress (Figure 5), without the formation of the m-phase. The compressive residual stresses gradually increased by the increase in aggressiveness of the sandblasting (larger particle size and higher pressure). Compressive residual stress in the surface layer tends to be advantageous because it tends to close cracks and slow crack propagation. Sandblasting the zirconia surface induces protective compressive residual stresses from the t–m phase transformation and, as a result, may increase the mechanical strength of the ceramic [12]. Gentle sandblasting (small-sized abrasive and low blasting pressure) may be beneficial because it induces limited damage, confined to the transformed region where a compressive stress field exists [11]. In some cases, zirconia ceramic may be strengthened by removing the weak grains and grinding traces using such gentle sandblasting treatment [19]. However, aggressive sandblasting (large abrasive particle and high blasting pressure) can weaken the ceramic by creating new surface flaws and microcracks which cannot be counteracted by the compressive residual stress field [11,12,19].

The combination of mechanical and chemical procedures is essential to the efficacy of the resin bonding procedure to zirconia [1]. Bisphenol A diglycidyl methacrylate (bis-GMA)-based resin material cannot form a durable adhesion to zirconia even after roughening by sandblasting [20]. High and durable bond strength can be achieved when an MDP-containing primer or an MDP-containing resin is applied to sandblasted zirconia [6,7]. It has been hypothesized that phosphate groups in the MDP molecule can potentially react with one or two zirconium atoms in zirconia crystals [6,21,22]. In this study, the MDP-containing resin luting material Panavia F 2.0 was used. Without sandblasting after polishing, however, the initial bonding (average 6.2 MPa) was reduced to an average 1.9 MPa and 0 MPa after 10,000 and 30,000 thermocycles, respectively (Table 2), with complete adhesive failure at the ceramic surface. These decreases may be attributable to the degradation of the resin luting material itself, the hydrolytic effect of water at the resin–zirconia interface, and mismatch between the thermal expansion coefficient between the resin and ceramic [1,23].

Sandblasting of zirconia increases the surface roughness and thus also the area for mechanical interlocking, resulting in acceptable micrometer scale roughness [1]. At the same time, the treatment cleans the surface by removing any organic contaminants, improves the wettability, and allows resin luting agent to flow into the roughened surface more effectively [7,24]. Rougher zirconia surfaces may provide wider contact areas and microporosities for resin luting agents [5]. Before thermocycling, the higher blasting pressure resulted in significantly higher SBS values within the same particle size condition (4B than 2B) (Table 2). However, such an increase in the surface roughness and residual stress, depending on the sandblasting conditions, did not significantly affect the SBS values after thermocycling (either 10,000 or 30,000 cycles). The CA data showed that more aggressive sandblasting did not necessarily increase the surface wettability, as seen in the roughness-corrected values, notwithstanding the more increased surface roughness values (Table 1).

Strong micromechanical interlocking can only be achieved when a resin luting material effectively penetrates and flows into the microretentions created on the sandblasted zirconia surface [25]. However, resin luting material may not completely fill the microporosities of the sandblasted surface due to its viscosity [26]. Thus, it is assumed that surface roughness exceeding the optimum for adequate micromechanical interlocking does not contribute to enhancement of resin adhesion to zirconia. Yang and Chang [13] demonstrated the relationship between bond strength and residual stress when HA coatings were plasma-sprayed on a titanium substrate. As resin adhered to zirconia seems less intimate at the interface as compared to HA coating to titanium, the influence of residual stress of zirconia on resin bonding may be minimal. However, a combined use of an adhesive primer, which could potentially more effectively wet the microporosities of the sandblasted zirconia surface due to its lower viscosity, and a bis-GMA resin cement was not tested in this study. Therefore, further investigation is needed to confirm this speculation.

In this study, the effect of four different sandblasting conditions (two blasting particle sizes and two pressures) on the resin bonding to zirconia was investigated in terms of residual stress. The residual stresses created on the zirconia surfaces directly depended on the aggressiveness of sandblasting. However, such increased surface roughness and residual stress did not affect the bonding durability of the adhesive resin cement with relatively high viscosity. Although sandblasting is favorable for improved resin bonding to zirconia, it can affect the mechanical properties of the ceramic due to accelerated t–m phase transformation, which may put stress on the surfaces and cause fatigue in the structure that could lead to premature and catastrophic failure [1]. Therefore, mild sandblasting of zirconia with small-sized abrasives and reduced blasting pressure is preferred to an aggressive procedure because increased surface roughness and residual stress do not directly affect the resin bonding durability. Future studies should include other resin cements with different viscosities to clarify the characteristics of resin–zirconia interface. It should also be noted that SBS test results tend to show high standard deviations due to uneven distribution of the stresses at the bonding interface [27]. Therefore, the use of other bond strength testing methodologies, such as the microtensile bond strength test, might be considered when evaluating resin bonding and its durability to zirconia ceramic.

## Figures and Tables

**Figure 1 materials-13-05629-f001:**
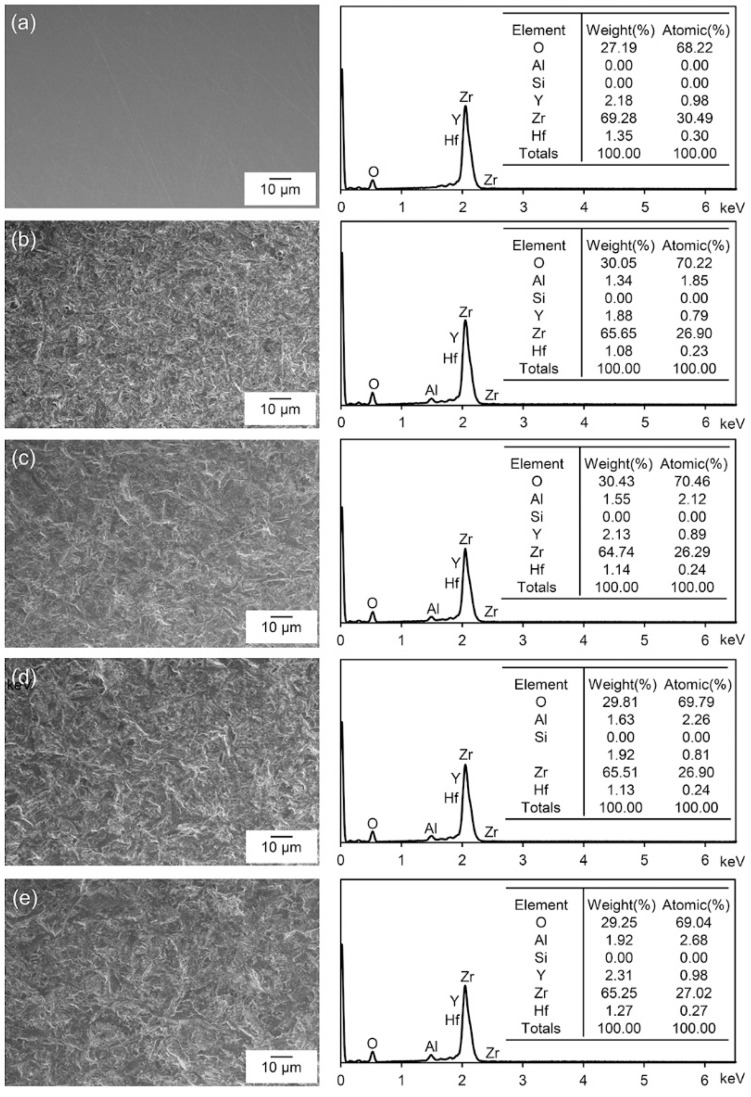
SEM images (1000×) of the polished (**a**) and sandblasted (**b**–**e**) zirconia surfaces and corresponding EDS spectra: (**b**) 50M-2B, (**c**) 50M-4B, (**d**) 110M-2B, and (**e**) 110M-4B.

**Figure 2 materials-13-05629-f002:**
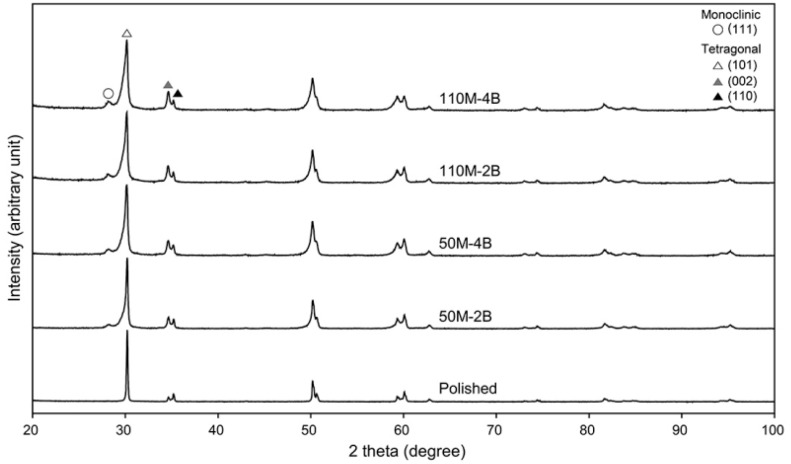
X-ray diffraction patterns for the polished and sandblasted zirconia surfaces.

**Figure 3 materials-13-05629-f003:**
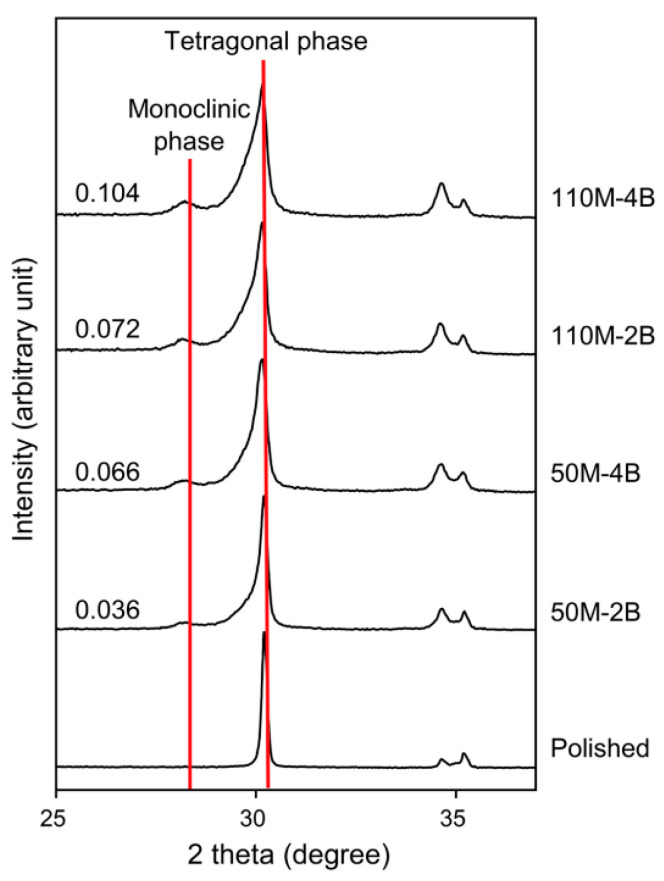
X-ray diffraction patterns for the polished and sandblasted zirconia surfaces, showing the increased monoclinic phase after sandblasting. The values on the left side of the graphs indicate the m-phase/(t-phase + m-phase) ratios. Monoclinic phase was not detected for the polished group.

**Figure 4 materials-13-05629-f004:**
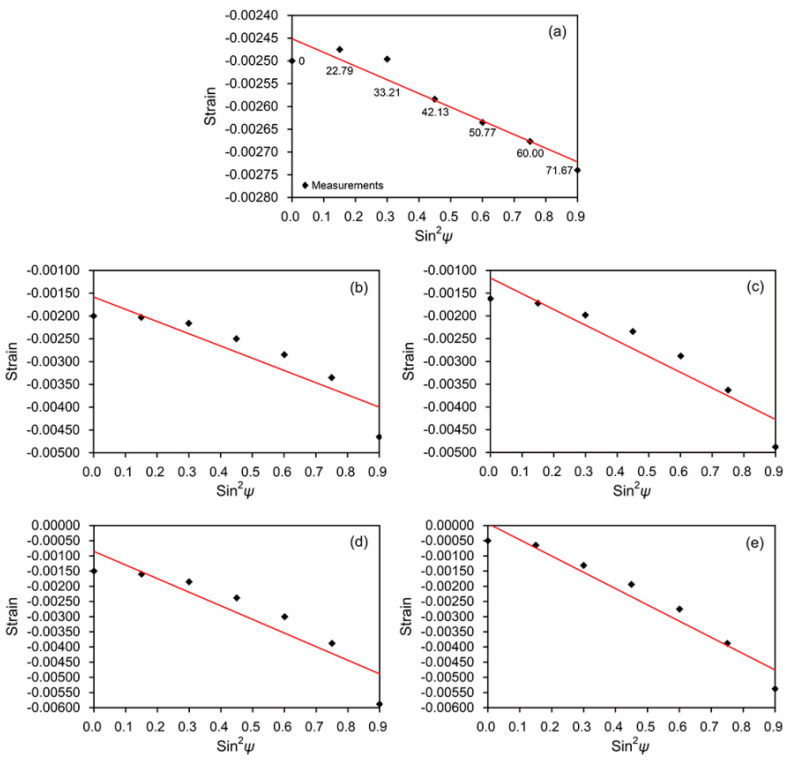
Linear dependence of the lattice strain (002) on sin^2^*ψ* for the polished (**a**) and sandblasted (**b**–**e**) zirconia surfaces: (**b**) 50M-2B, (**c**) 50M-4B, (**d**) 110M-2B, and (**e**) 110M-4B.

**Figure 5 materials-13-05629-f005:**
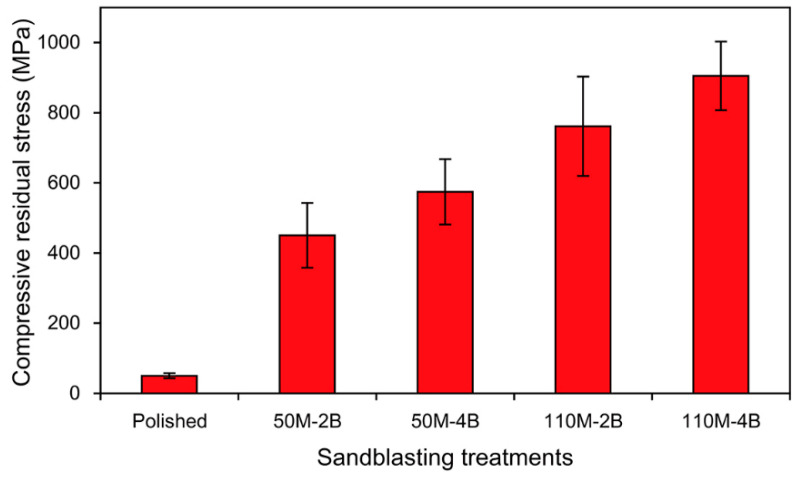
Compressive residual stresses for the polished and sandblasted zirconia surfaces.

**Table 1 materials-13-05629-t001:** Mean (standard deviation) surface roughness values and water contact angles for the polished and sandblasted groups.

Groups	Surface Roughness (μm)		Water Contact Angle (Degrees)	
	*R* _a_	*R* _z_	Measured	Roughness-Corrected
Polished	0.04 (0.003) ^a,1^	0.29 (0.03) ^a^	59.7 (3.2) ^a^	60.7 (3.1) ^a^
50M-2B	0.63 (0.03) ^b^	4.52 (0.17) ^b^	44.1 (1.3) ^b^	49.2 (1.1) ^b^
50M-4B	0.75 (0.04) ^c^	5.16 (0.20) ^c^	43.5 (1.4) ^b^	49.3 (1.2) ^b^
110M-2B	0.84 (0.03) ^d^	6.01 (0.28) ^d^	42.4 (1.7) ^b,c^	48.7 (1.4) ^b^
110M-4B	1.06 (0.07) ^e^	7.53 (0.26) ^e^	40.9 (1.2) ^c^	48.5 (0.9) ^b^

^1^ Within the same column, the same lowercase superscript letters ^a–e^ indicate no significant differences among the groups (*p* > 0.05).

**Table 2 materials-13-05629-t002:** Shear bond strength (in MPa) of the resin cement to the polished and sandblasted zirconia ceramic surfaces (*n* = 10).

Groups	0 Thermocycles (1-Day Water Immersion)	10,000 Thermocycles (after 7-Day Water Immersion)	30,000 Thermocycles (after 7-Day Water Immersion)
Polished	6.2 (1.4) ^A,a,1^	1.9 (0.7) ^A,b^	0.0 (0.0) ^A,c^
50M-2B	16.9 (3.3) ^B,a,2^	14.6 (2.2) ^B,a^	11.5 (3.0) ^B,b^
50M-4B	20.8 (2.7) ^C,a^	17.2 (2.9) ^B,b^	11.1 (2.9) ^B,c^
110M-2B	17.3 (2.5) ^B,a^	15.1 (2.0) ^B,a^	10.4 (2.3) ^B,b^
110M-4B	21.4 (2.6) ^C,a^	16.4 (1.9) ^B,b^	10.5 (2.5) ^B,c^

^1^ Within the same column, the same uppercase superscript letters ^A–C^ indicate no significant differences among the groups (*p* > 0.05). ^2^ Within the same row, the same lowercase superscript letters ^a–c^ indicate no significant differences among the thermocycling conditions (*p* > 0.05).

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
