# Peer review of "Influence of Sandblasting Particle Size and Pressure on Resin Bonding Durability to Zirconia: A Residual Stress Study"

_materials, 2020, doi:10.3390/ma13245629_

Round 1

Reviewer 1 Report

The manuscript falls within the scope of adhesion of zirconia, a topic of current interest, not yet fully clarified, since we currently do not have a standardized adhesion protocol available. The work is well organized and examines, practical aspects in the execution of sandblasting. There are already data in the literature regarding the influence of sandblasting parameters on bond strength, surface characteristics and phase change. The results support and supplement the indications already provided by other authors. This study is in fact supported by comprehensive laboratory documentation.

In my opinion some aspects of the materials and methods should be deepened, aligning the discussion with the recommendations.

Describe more precisely the total sample size, the number of samples that are used for the various tests, and whether they are then used for further tests. It would be appropriate to describe according to which principle the sample size was defined.

The authors arbitrarily choose the parameters related to particle size and pressure. It would be advisable to point out the reason of this choice, as some authors recommend using even less aggressive parameters to perform sandblasting.

Introduction.

Line 36. “This ceramic is in popular use due to its superior properties, such as high flexural strength, high fracture toughness, esthetics, and superior biocompatibility [1,2]”  The sentence should be changed, zirconia is not superior to other ceramics regarding all the listed characteristics.

Zhang Y, Lawn BR. Novel Zirconia Materials in Dentistry. J Dent Res. 2018 Feb;97(2):140-147.

Line 47. As also stated by the authors, to date there is no safe and standardized protocol for the adhesion of zirconia. It is probably an overestimate to call it "high and durable".

Materials and Methods.

Line 77.Indicate the ultrasonic cleaning times, it is important since the usefulness of this treatment and the permanence of alumina residues are then discussed.

Line 120. It could be more correct to describe the protocol accurately than to assert “The experimental protocols have been described elsewhere”.

Discussion

Line 254. You stated It has been reported that phosphate groups in the MDP molecule can react with one or two zirconium atoms in zirconia crystals”. Although citations have been reported in reference to this assertion, in my opinion it is not totally correct. The way in which 10-MDP binds to zirconia has not yet been fully clarified, studies report "thermodynamically favourable" or "possible" configurations, and it is not clear whether the zirconia atom is directly involved in the bond.

Xie H, Tay FR, Zhang F, Lu Y, Shen S, Chen C. Coupling of 10-methacryloyloxydecyldihydrogenphosphate to tetragonal zirconia: Effect of pH reaction conditions on coordinate bonding. Dent Mater. 2015 Oct;31(10):e218-25.

Nagaoka N, Yoshihara K, Feitosa VP, Tamada Y, Irie M, Yoshida Y, Van Meerbeek B, Hayakawa S. Chemical interaction mechanism of 10-MDP with zirconia. Sci Rep. 2017 Mar 30;7:45563. doi: 10.1038/srep45563.

Line 276. “It is assumed that surface roughness exceeding the optimum for adequate micromechanical interlocking does not contribute to enhancement of resin adhesion to zirconia.”This assumption can only be made in the case of a protocol that uses resinous cement and does not foresee the use for example of more fluid primers. The authors mention in the conclusions that the viscosity of the materials used can affect the adhesion and the possibility of exploiting greater surface roughness. This should be deepened in the discussion.

Author Response

The manuscript falls within the scope of adhesion of zirconia, a topic of current interest, not yet fully clarified, since we currently do not have a standardized adhesion protocol available. The work is well organized and examines, practical aspects in the execution of sandblasting. There are already data in the literature regarding the influence of sandblasting parameters on bond strength, surface characteristics and phase change. The results support and supplement the indications already provided by other authors. This study is in fact supported by comprehensive laboratory documentation.

- We agree with your comment that there are already numerous studies in the literature regarding the influence of sandblasting parameters on bond strength, surface characteristics and phase change. Nonetheless, we focused on the tetragonal-to-monoclinic phase transformation and residual stresses induced by sandblasting in the present study.

In my opinion some aspects of the materials and methods should be deepened, aligning the discussion with the recommendations.

- Thank you for your suggestion. We have made every effort to enhance the quality of our manuscript by revising our draft in response to your comments.

Describe more precisely the total sample size, the number of samples that are used for the various tests, and whether they are then used for further tests. It would be appropriate to describe according to which principle the sample size was defined.

- In this revision, the number of samples for each group has been more clarified. In our study, the surface roughness, contact angle, and shear bond strength data were subjected to statistical analysis. Although the sample size was arbitrarily determined without any power analysis based on the earlier studies, the significant differences among the groups in the values (Tables 1 and 2) indicate that the sample size in our study was sufficient to minimize the risk of committing a type II error.

The authors arbitrarily choose the parameters related to particle size and pressure. It would be advisable to point out the reason of this choice, as some authors recommend using even less aggressive parameters to perform sandblasting.

- Thank you for your comment. The two particle sizes and the two blasting pressures were determined based on the earlier studies (Hallmann, L.; Ulmer, P.; Reusser, E.; Hämmerle, C.H.F. Surf. Coat. Technol. 2012, 206, 4293–4302; Chintapalli, R.K.; Rodriguez, A.M.; Marro, F.G.; Anglada, M. J. Mech. Behav. Biomed. Mater. 2014, 29, 126–137) (Page 2, Lines 77–80). Although a less aggressive sandblasting condition (e.g., 30 μm alumina, 0.05 MPa) can be recommended, the combination of 50 μm alumina and 0.2 MPa might be considered as a gentile sandblasting condition.

Introduction.

Line 36. “This ceramic is in popular use due to its superior properties, such as high flexural strength, high fracture toughness, esthetics, and superior biocompatibility [1,2]”  The sentence should be changed, zirconia is not superior to other ceramics regarding all the listed characteristics.

Zhang Y, Lawn BR. Novel Zirconia Materials in Dentistry. J Dent Res. 2018 Feb;97(2):140-147.

- Thank you for introducing the excellent review paper. The sentence has been modified by referring to the paper (Page 1, Lines 35–38).

Line 47. As also stated by the authors, to date there is no safe and standardized protocol for the adhesion of zirconia. It is probably an overestimate to call it "high and durable".

- We agree with your comment. The sentence has been slightly modified (Page 2, Lines 43–45).

Materials and Methods.

Line 77.Indicate the ultrasonic cleaning times, it is important since the usefulness of this treatment and the permanence of alumina residues are then discussed.

- Thank you for pointing it out. The ultrasonic cleaning times have been addressed following your suggestion (Page 2, Lines 82–83).

Line 120. It could be more correct to describe the protocol accurately than to assert “The experimental protocols have been described elsewhere”.

- We agree with your suggestion. The sentence “The experimental protocols have been described elsewhere.” has been removed and more detailed bonding and debonding procedures have been addressed.

Discussion

Line 254. You stated “It has been reported that phosphate groups in the MDP molecule can react with one or two zirconium atoms in zirconia crystals”. Although citations have been reported in reference to this assertion, in my opinion it is not totally correct. The way in which 10-MDP binds to zirconia has not yet been fully clarified, studies report "thermodynamically favourable" or "possible" configurations, and it is not clear whether the zirconia atom is directly involved in the bond.

Xie H, Tay FR, Zhang F, Lu Y, Shen S, Chen C. Coupling of 10-methacryloyloxydecyldihydrogenphosphate to tetragonal zirconia: Effect of pH reaction conditions on coordinate bonding. Dent Mater. 2015 Oct;31(10):e218-25.

Nagaoka N, Yoshihara K, Feitosa VP, Tamada Y, Irie M, Yoshida Y, Van Meerbeek B, Hayakawa S. Chemical interaction mechanism of 10-MDP with zirconia. Sci Rep. 2017 Mar 30;7:45563. doi: 10.1038/srep45563.

- We agree with your comment. The sentence has been slightly modified by incorporating the paper of Nagaoka et al. (2017) (Page 10, Lines 266–267).

Line 276. “It is assumed that surface roughness exceeding the optimum for adequate micromechanical interlocking does not contribute to enhancement of resin adhesion to zirconia.” This assumption can only be made in the case of a protocol that uses resinous cement and does not foresee the use for example of more fluid primers. The authors mention in the conclusions that the viscosity of the materials used can affect the adhesion and the possibility of exploiting greater surface roughness. This should be deepened in the discussion.

- Thank you for pointing it out. We strongly agree with your opinion. A new sentence has been added to clarify that the speculation needs further investigations (Page 11, Lines 293–296).

Reviewer 2 Report

The paper approaches a topic of interest, i.e. the influence of a surface treatment method (sandblasting) on resin bonding strength to zirconia. In general, the study is well-done and the manuscript is well written, therefore the paper could be considered for publication in Materials, with some improvements, as pointed out in the following.

1) The English is good, but it could still be polished. Please check the manuscript. Abstract could be written at the present tense. Refs should be always placed before a punctuation sign, etc.

2) The Abstract could be completed, maybe with less numbers but with more background and conclusions.

3) Provided refs are fine, but considering how intense this topic has been studied, there should be much more refs, including for example

-on zirconia properties

Craciunescu E., Sinescu C., Negrutiu M.L., Pop D.M., Lauer H.-C., Rominu M., Hutiu Gh., Bunoiu M., Duma V.-F., Antoniac I., “Shear Bond Strength Tests of Zirconia Veneering Ceramics after Chipping Repair”, Journal of Adhesion Science and Technology 30(6), 666-676, 2016.

-on SBS testing

Dundar M, Ozcan M, Gokce B, et al. Comparison of two bond strength testing methodologies

for bilayered all-ceramics. Dent. Mater. 2007; 23: 630–636.

-on other methods for surface processing, including for Discussion, please see comments 6, 7.

4) Figure 1 must be corrected, with larger inscriptions, they cannot be red now. Please check all figs (see for example 4 and 5) to provide equal and large enough inscriptions in all of them.

One can also have difficulties in seeing a difference between SEM images in Fig. 1. Scales are also not visible.

5) Why point out "in vitro" - line 64? Could one do sandblasting in vivo?

6) The authors focus on one method to process the surface. What about other methods? One must carefully explain from the Intro the choice of (only) one method and ignoring others. 

7) Also, at Discussions comparisons with results obtained with other methods must be provided, in quite an extensive discussion. Readers must be convinced on relevance and efficiency of method & study. This is missing now.

8) Once an acronym is introduced, e.g. SBS, please be consistent and use it throughout the entire manuscript. please check all such acronyms.

9) Conclusions section is much too brief. It could be included in the same section with Discussion.

10) The Discussion would be nice to be more structured, therefore easier to follow by readers, using some points/bullets. Ideas could also be more organized.  

Author Response

The paper approaches a topic of interest, i.e. the influence of a surface treatment method (sandblasting) on resin bonding strength to zirconia. In general, the study is well-done and the manuscript is well written, therefore the paper could be considered for publication in Materials, with some improvements, as pointed out in the following.

- Thank you very much for your encouraging comment.

1) The English is good, but it could still be polished. Please check the manuscript. Abstract could be written at the present tense. Refs should be always placed before a punctuation sign, etc.

- According to your suggestions, we have checked and revised throughout the manuscript, including correcting errors and typos.

2) The Abstract could be completed, maybe with less numbers but with more background and conclusions.

- We have revised the Abstract section, including adding the background of our study.

3) Provided refs are fine, but considering how intense this topic has been studied, there should be much more refs, including for example

-on zirconia properties

Craciunescu E., Sinescu C., Negrutiu M.L., Pop D.M., Lauer H.-C., Rominu M., Hutiu Gh., Bunoiu M., Duma V.-F., Antoniac I., “Shear Bond Strength Tests of Zirconia Veneering Ceramics after Chipping Repair”, Journal of Adhesion Science and Technology 30(6), 666-676, 2016.

-on SBS testing

Dundar M, Ozcan M, Gokce B, et al. Comparison of two bond strength testing methodologies for bilayered all-ceramics. Dent. Mater. 2007; 23: 630–636.

-on other methods for surface processing, including for Discussion, please see comments 6, 7.

- Thank you for introducing the excellent papers. We have incorporated the articles and other papers as new references and discussed in this revision.

4) Figure 1 must be corrected, with larger inscriptions, they cannot be red now. Please check all figs (see for example 4 and 5) to provide equal and large enough inscriptions in all of them.

- Thank you for pointing it out. We have checked all figures and corrected some of them if necessary. Figure 1 has been corrected to ensure clarity and readability. Figures 4 and 5 have also been enlarged in the main text.

One can also have difficulties in seeing a difference between SEM images in Fig. 1. Scales are also not visible.

- The SEM images have been enlarged in the main text. The scale bars in the images have also been modified.

5) Why point out "in vitro" - line 64? Could one do sandblasting in vivo?

- The inappropriate term “in vitro” has been removed (Page 2, Line 69). Thank you.

6) The authors focus on one method to process the surface. What about other methods? One must carefully explain from the Intro the choice of (only) one method and ignoring others. 

- Thank you pointing it out. In this study, we only focused on the zirconia surface roughening by sandblasting with alumina particles. Another important roughening method (tribochemical silica-coating) should have been addressed in our draft. In this revision, we have discussed the merit and problems of the treatment with new references.

7) Also, at Discussions comparisons with results obtained with other methods must be provided, in quite an extensive discussion. Readers must be convinced on relevance and efficiency of method & study. This is missing now.

- We agree with your comment. Sandblasting conditions using silica-coated alumina particles were not included in this study. This topic has been addressed for comparison purpose in the Discussion section.

8) Once an acronym is introduced, e.g. SBS, please be consistent and use it throughout the entire manuscript. please check all such acronyms.

- We have checked and corrected the use of all acronyms throughout the manuscript. Thank you.

9) Conclusions section is much too brief. It could be included in the same section with Discussion.

- Following your suggestion, the last paragraph of the Discussion section has been revised by incorporating the sentences of the Conclusions section.

10) The Discussion would be nice to be more structured, therefore easier to follow by readers, using some points/bullets. Ideas could also be more organized.  

- We have made every effort to revise the Discussion section for readers’ easier reading. If additional changes and corrections still may be necessary, please allow us to revise our paper further. Once again, thank you very much for all your valuable comments and suggestions.

Round 2

Reviewer 2 Report

The manuscript has been corrected and completed according to all the comments made. In the opinion of this reader, it can be considered for publication in the present form.